# Comparative Transcriptome Analysis Reveals a Potential Regulatory Network for Ogura Cytoplasmic Male Sterility in Cabbage (*Brassica oleracea* L.)

**DOI:** 10.3390/ijms24076703

**Published:** 2023-04-04

**Authors:** Li Chen, Wenjing Ren, Bin Zhang, Huiling Guo, Zhiyuan Fang, Limei Yang, Mu Zhuang, Honghao Lv, Yong Wang, Jialei Ji, Xilin Hou, Yangyong Zhang

**Affiliations:** 1State Key Laboratory of Crop Genetics and Germplasm Enhancement, College of Horticulture, Nanjing Agricultural University, Nanjing 210095, China; 18205480752@163.com (L.C.); 17863805323@163.com (W.R.); 13126720352@163.com (B.Z.); 2State Key Laboratory of Vegetable Biobreeding, Institute of Vegetables and Flowers, Chinese Academy of Agricultural Sciences, Beijing 100081, China; caas_ivf_guohl@163.com (H.G.); fangzhiyuan@caas.cn (Z.F.); yanglimei@caas.cn (L.Y.); zhuangmu@caas.cn (M.Z.); lvhonghao@caas.cn (H.L.); wangyong@caas.cn (Y.W.); jijialei@caas.cn (J.J.)

**Keywords:** cabbage, Ogura CMS, pollen development, comparative transcriptome analysis

## Abstract

Ogura cytoplasmic male sterility (CMS) lines are widely used breeding materials in cruciferous crops and play important roles in heterosis utilization; however, the sterility mechanism remains unclear. To investigate the microspore development process and gene expression changes after the introduction of *orf138* and *Rfo*, cytological observation and transcriptome analysis were performed using a maintainer line, an Ogura CMS line, and a restorer line. Semithin sections of microspores at different developmental stages showed that the degradation of tapetal cells began at the tetrad stage in the Ogura CMS line, while it occurred at the bicellular microspore stage to the tricellular microspore stage in the maintainer and restorer lines. Therefore, early degradation of tapetal cells may be the cause of pollen abortion. Transcriptome analysis results showed that a total of 1287 DEGs had consistent expression trends in the maintainer line and restorer line, but were significantly up- or down-regulated in the Ogura CMS line, indicating that they may be closely related to pollen abortion. Functional annotation showed that the 1287 core DEGs included a large number of genes related to pollen development, oxidative phosphorylation, carbohydrate, lipid, and protein metabolism. In addition, further verification elucidated that down-regulated expression of genes related to energy metabolism led to decreased ATP content and excessive ROS accumulation in the anthers of Ogura CMS. Based on these results, we propose a transcriptome-mediated induction and regulatory network for cabbage Ogura CMS. Our research provides new insights into the mechanism of pollen abortion and fertility restoration in Ogura CMS.

## 1. Introduction

Cabbage is an important cruciferous vegetable crop that is enjoyed by people worldwide for its crisp taste and rich nutritional value. According to FAO statistics, the global cabbage planting area was 2.45 million hectares in 2021, with a total yield of 71.7 million tons [1]. Cabbage is a typical cross-pollinated plant, and a consensus decision has been made among breeders to use heterosis to improve its yield and quality [2,3,4,5]. Cytoplasmic male sterility (CMS) is a maternal genetic trait that is widely found in higher plants [6], and its one of the main tools for heterosis utilization in cruciferous crops [7]. CMS plants have normal vegetative growth and pistil development, but cannot produce functional pollen, which has a natural advantage in hybrid production [8]. In addition, CMS plants are ideal materials for studying pollen development and nucleo–cytoplasmic interactions [9,10]. At present, Ogura CMS is the most widely used CMS type in cabbage hybrid production. It has the advantages of stable sterility and being easy transfer, and its fertility can be restored by the nuclear gene *Rfo* [7,11]. Ogura CMS was first discovered in wild radish (*Raphanus sativus*) [12]. Since then, breeders have gradually transferred the CMS trait and the restorer gene *Rfo* to cabbage through intergeneric hybridization and protoplast fusion [13,14,15], and created many new excellent cabbage varieties using this characteristic [5,16].

CMS was first discovered by Kölreuter in intraspecific and interspecific hybrids [17]. Breeders subsequently found that a higher proportion of male sterile material was produced in self-crossing or hybrid offspring [18,19]. In 1976, it was proven for the first time that mitochondrial DNA (mtDNA) was the carrier of CMS-related genes through restriction fragment polymorphism [20]. Studies have found that plant mitochondrial genomes are highly variable in both size and structure and contain a large number of repeated sequences [21]. During self-crossing and hybridization, frequent recombination of mitochondrial genomes leads to the formation of new open reading frames and substoichiometric changes [22]. These open reading frames usually form chimeric structures with known mitochondrial genes, resulting in new gene expression patterns and even novel genes, including those that cause CMS [23]. Most CMS genes form chimeras with ATP synthase subunits or cytochrome oxidase subunits and encode proteins with transmembrane domains, such as *urf13* and *atp4* in maize T-CMS [24], *orfH79* and *atp6* in rice HL-CMS [25], *orf522* and *atp1* in sunflower PET1-CMS [26], *orf507* and *cox2* in pepper Peterson-CMS [27], *orf224* and *atp6* in *Brassica napus* pol-CMS [28], and *orf138* and *atp8* in radish Ogura CMS [29]. These chimeric genes may disrupt mitochondrial function during the critical period of microspore formation, causing the abnormal development of floral organs or pollen grains, eventually leading to CMS [30].

The complex regulatory system of pollen development and the particularity of mitochondrial genes have caused great confusion in the study of CMS mechanisms. In recent years, with the development of high-throughput sequencing technology and bioinformatics, an increasing number of researchers have explored the molecular mechanism of CMS through comparative analysis of the transcriptomes, proteomes, or metabolomes. A combined analysis of the transcriptomes and proteomes between wheat AL-CMS and its maintainers revealed that most of the key regulatory genes related to anther development were enzymes or transcription factors (TFs) [31]. In addition, a transcriptome analysis of wheat K-TCMS showed that genes related to phenylpropane biosynthesis and jasmonic acid biosynthesis were upregulated in CMS anthers and regulated by MYB TFs [32]. Yang et al. found that 1716 differentially expressed genes (DEGs) related to pollen development in eggplant are involved mainly in redox reactions, carbohydrate and amino acid metabolism, and transcriptional regulation [33]. In onion, two cytoplasmic genes, *atp9* and *cox1*, and three nuclear genes, *SERK1*, *AG,* and *AMS,* play important roles in pollen development according to an analysis of DEGs [34]. In a study on *B. napus*, RNA-seq data of Pol CMS and its maintainer flower buds showed that 1148 unigenes had significant differences in expression and that these unigenes were involved mainly in metabolic and protein synthesis pathways [35]. Tobacco *sua*-CMS is the only CMS system in tobacco breeding. A comparative transcriptomic analysis showed that genes involved in the ER stress pathway, heat shock protein family, F_1_F_0_-ATPase, and differentiation of stages are down-regulated in *sua*-CMS, while genes in the programmed cell death (PCD) pathway are significantly up-regulated, indicating that PCD and defects in ATP synthesis are crucial to the pollen abortion of *sua*-CMS [36]. A proteomic analysis of Shaan2A-CMS in *B. napus* showed that the differentially expressed proteins were related mainly to carbohydrate metabolism, energy metabolism, and genetic information processing pathways [37]. In rice ZD-CMS, a large number of proteins involved in carbohydrate metabolism or stress response are down-regulated, indicating that these metabolic processes are hindered during pollen development [38].

In the present study, we observed the microspore development process of a maintainer line, Ogura CMS line, and restorer line by semithin sectioning and determined the pollen abortion period in Ogura CMS. The gene expression profiles of the three lines were obtained by full-length transcriptome sequencing, and the genes specifically expressed in the Ogura CMS line were identified. The transcriptional regulatory network of Ogura CMS was constructed by functional annotation and analysis. These findings lay a foundation for further study of the pollen development process and CMS mechanism.

## 2. Results

### 2.1. Morphological and Cytological Comparison between the Maintainer Line, Ogura CMS Line, and Restorer Line

Anatomical observation of flowers in the same period showed that the stamens of maintainer and restorer flowers were higher than those of pistil, and the surfaces of all stamens were covered with yellow pollen grains (Figure 1A,C), while the stamens of Ogura CMS flowers were lower than those of the pistil, and the anthers were withered without pollen grains attached to the surface (Figure 1B). Alexander staining showed that the anthers of Ogura CMS did not contain viable pollen (Figure 1E).

The male gametophytes at different developmental stages in the three lines were observed on semithin sections with toluidine blue staining (Figure 2). The results showed that there were no significant cytological differences among the three lines from the sporogenesis stage to the meiosis stage. The tapetal cells in the Ogura CMS lines were severely deformed and separated from the pollen sac walls to squeeze the microspores at the tetrad stage, and the cell walls of some microspores were degraded and showed irregular shapes (Figure 2I). From the uninucleate stage to the trinucleate stage, the tapetal cells in the Ogura CMS line were further degraded, showing high vacuolation and irregularity, and continued to squeeze the microspores inward. At the same time, pollen wall rupture, microspore development arrest, and gradual degradation were observed. By the mature pollen stage, the tapetum and microspores in the Ogura CMS line were completely degraded, and no pollen grains formed in the pollen sac (Figure 2L). In the maintainer line and restorer line, the initial degradation of tapetal cells occurred in the bicellular micropore stage to the trinucleate micropore stage (Figure 2E,Q), and tapetal cells remained until the mature pollen stage (Figure 2F,R). In conclusion, the early degradation of tapetal cells is the main reason for the failure of Ogura CMS to produce mature pollen.

### 2.2. Transcriptome Sequencing and Assembly

In this study, full-length transcriptome sequencing was performed on the maintainer line 19–616, the Ogura CMS line 19–2167, and the restorer line FR2202 using the ONT third-generation sequencing platform, and 5.28 Gb, 7.05 Gb, and 4.90 Gb of data were obtained, respectively (Table 1). After filtering out low-quality reads and removing adapters, clean data were obtained. A total of 7,557,374 clean reads were obtained from the maintainer line 19–616, which produced 6,429,226 full-length sequences. The N50 was 897 nt. A total of 10,657,713 clean reads were obtained from the Ogura CMS line 19–2167, from which 9,214,366 full-length sequences were obtained, and the N50 was 857 nt. For the restorer line FR2202, a total of 7,244,891 clean reads were obtained, from which 6,313,210 full-length sequences were obtained, and the N50 was 841 nt. The MeanQscore of all samples was greater than 10.3, indicating that the sequencing data were of high quality and could be used for subsequent analysis.

### 2.3. Differential Expression Analysis of Nuclear Genes

The identified full-length sequences were aligned to the reference genome of *B. oleracea* (https://plants.ensembl.org/Brassica_oleracea/Info/Index (accessed on 3 June 2020)) [39] to obtain the nuclear gene transcripts. The maintainer line 19–616, the Ogura CMS line 19–2167, and the restorer line FR2202 were compared and analyzed in pairs. In total, 2505 DEGs were identified in Ogura CMS 19–2167 compared with the maintainer line 19–616, of which 1491 genes were significantly down-regulated and 1014 genes were significantly up-regulated (Figure 3A). A total of 1986 DEGs were identified in the restorer line FR2202, of which 556 genes were down-regulated and 1430 genes were up-regulated (Figure 3B). Compared with Ogura CMS 19–2167, the restorer line FR2202 exhibited 3948 DEGs, among which 1158 genes were down-regulated and 2790 genes were up-regulated (Figure 3C).

To explore the molecular mechanisms related to the pollen abortion of Ogura CMS, we screened DEGs by analyzing the shared genes between the three lines. According to the overlap between the comparison groups, the whole Venn diagram was divided into seven parts (I–VII), and the number and proportion of DEGs identified in each part are shown in Figure 3D. Among them, the DEGs in part IV represent those genes that were significantly differentially expressed in the CMS line compared with the maintainer line, but whose expression levels returned to normal in the restorer line, indicating that they may respond to the regulation of CMS factors at the transcriptional or post-transcriptional level. Therefore, we believe that the DEGs in part IV are the most closely related genes to the pollen abortion of Ogura CMS. A total of 1287 DEGs were included in part IV, of which 884 DEGs were down-regulated and 403 DEGs were up-regulated in the Ogura CMS line, respectively. We validated the accuracy of the RNA-seq data by qRT–PCR analysis (Appendix A).

### 2.4. Functional Annotation and Enrichment Analysis of DEGs

GO enrichment analysis was conducted to reveal significantly enriched GO terms of the 1287 core DEGs in part IV. A total of 883 DEGs were significantly enriched with 137 GO terms (*p*-value < 0.05) in the three categories: biological process (BP), cellular component (CC), and molecular function (MF) (Figure 4A, Appendix A).

In the BP category, a total of 368 DEGs were enriched with 67 GO terms associated with the lipid catalytic process (GO: 0016042), leaf sensitivity (GO: 0010150), specific catalytic process (GO: 0045490), protein involved in cellular protein catalytic process (GO: 0051603), killing of cells of other organization (GO: 0031640), etc. In the MF category, 342 DEGs were enriched with 56 GO terms, including lipid binding (GO: 0008289), oxidoreductase activity (GO: 0016491), lipase activity (GO: 0016298), proteinase activity (GO: 0030599), and cysteine-type endopeptidase activity (GO: 0004197). In the CC category, 173 DEGs were enriched with 14 GO terms, such as anchored component of membrane (GO: 0031225), monolayer surrounded lipid storage body (GO: 0012511), and pollen coat (GO: 0070505). The expression trends of some core DEGs among the three lines are shown in Figure 4B.

### 2.5. DEGs Involved in Pollen Development

Among the 1287 core DEGs, we identified a large number of genes involved in pollen development (Figure 5A, Appendix A). For example, two genes encoding the cellulose synthase A subunit and two genes encoding cinnamyl-CoA reductase 1 involved in cellulose and lignin biosynthesis were down-regulated in the Ogura CMS line compared with the maintainer line and restorer lines. Seven pectinesterase genes and four pectinesterase inhibitor genes mediating pectin demethylesterification were significantly down-regulated and up-regulated, respectively. Seven genes involved in phenylpropanoid biosynthesis were significantly down-regulated, including two caffeoyl-CoA O-methyltransferase genes, two spermidine hydroxycinnamoyl transferase genes, one chalcone hydroxone isomerase gene, and one type III polyketide synthase C gene. With regard to pollen wall biosynthesis, 21 genes were significantly down-regulated, including eight genes encoding GDSL esterase/lipase, eight genes encoding polygalacturonase (PG), three genes encoding the anther-specific protein BP4C and two genes encoding pollen coat protein. In addition, four pollen-specific protein genes, three anther-specific proline-rich protein genes, and one callose synthase gene that play important roles in the development of the tapetum and microspores were significantly down-regulated in the Ogura CMS line.

### 2.6. DEGs Involved in Energy Metabolism

Glycolysis, the tricarboxylic acid cycle, and oxidative phosphorylation are the three major pathways of ATP synthesis in plants and are essential for the normal development of plants [40]. We found that a total of 20 genes involved in the electron transport chain were significantly down-regulated in the Ogura CMS line compared with the maintainer line and the restorer line, including seven genes encoding the ATPase subunit, four genes encoding the NADH dehydrogenase α subcomplex subunit, four cytochrome P450 family genes, two genes encoding the cytochrome c oxidase subunit, two pyrophosphatase genes, and one ADP/ATP carrier protein gene. In addition, two genes encoding phosphate 3-phosphate dehydrogenase, which is the key enzyme in the glycolysis pathway, and genes encoding citrate synthase and the pyruvate dehydrogenase E1 subunit involved in the tricarboxylic acid cycle were also significantly down-regulated in Ogura CMS (Figure 5B, Appendix A).

Copper plays an important role in redox reactions due to its ability to accept and provide electrons [41]. Three copper transport protein genes and one blue copper protein gene were significantly down-regulated in Ogura CMS. Two thioredoxin genes and one catalase gene, which are involved in redox reactions, were also down-regulated.

### 2.7. DEGs Involved in Carbohydrate, Lipid, and Protein Metabolism

Carbohydrates are necessary for the development of microspores [42]. A large number of genes involved in carbon metabolism were significantly down-regulated in Ogura CMS, including 10 arabinogalactan protein genes, five fasciclin-like arabinogalactan protein genes, two beta-1,3-galactosyltransferase genes, one formate-tetrahydrofolate ligase gene, one metallenetetrahydrolase reductase gene, one UDP-arabinose 4-epimerase 3 gene, one ADP-ribosylation factor gene, and one eglucan endo-1,3-beta-glucosidase gene. For starch and sucrose metabolism, 18 genes were significantly down-regulated, including seven sugar transport protein genes, four beta-glucosidase genes, three beta glucosidase genes, one sugar transporter ERD6-like gene, one glycosyltransferase gene, one fructokinase gene, and one gene encoding glucose-1-phosphate adenylyltransferase large subunit (Figure 5C, Appendix A).

Lipid metabolism plays an important role in tapetal development and pollen wall formation [42]. Forty-four genes involved in lipid metabolism were significantly down-regulated in Ogura CMS, including long chain acyl-CoA synthetase 6,3-ketoacyl-CoA synthase and palmitoyl-protein thioesterase 1, which are involved in fatty acid metabolism; phosphoethanolamine N-methyltransferase, diacylglycerol kinase, ethanolamine-phosphate cytidylyltransferase, and phospholipase D alpha 2, which are involved in glycerophospholipid metabolism; delta-24-sterol reductase, 3-beta-hydroxysteroid-delta (8)-delta (7)-isomerase, and methylsterol monooxygenase, which are involved in steroid biosynthesis; jasmonate o-methyltransferase, which is involved in alpha-linolenic acid metabolism; and leucine aminopeptidase, which is involved in arachidonic acid metabolism. In addition, 17 oleosin genes and 10 non-specific lipid-transfer protein-like genes were down-regulated in Ogura CMS (Figure 5C, Appendix A).

Previous studies have shown that amino acids may be closely related to anther development and pollen production [43]. We observed changes in the expression of many genes involved in amino acid and protein metabolism. Among them, 46 genes were down-regulated in Ogura CMS, including three aminopeptidase genes and two glutathione S-transferase genes involved in glutathione metabolism; S-adenosylmethionine synthase, which is involved in cysteine and methionine metabolism; the adenylyl-sulfate kinase, which is involved in purine metabolism; glutamine synthetase, which is involved in nitrogen metabolism; S-adenosylmethionine decarboxylase, which is involved in arginine and proline metabolism; diaminopimelate decarboxylase 2 and lysine methyltransferase 2, which are involved in lysine biosynthesis; histidine biosynthesis bifunctional protein gene, which is involved in histidine metabolism; five cysteine proteinase inhibitor genes and one cell division control protein gene, which are involved in protein processing; and 28 ribosomal protein genes involved in ribosome biogenesis (Figure 5C, Appendix A).

Twelve genes involved in protein degradation were up-regulated in Ogura CMS, including six chaperone protein dnaJ genes, two Ring-H2 finger protein genes, one chaperonin 60 subunit gene, one branched-chain amino acid aminotransferase 2 gene, one methylcrotonoyl-CoA carboxylase gene, and one methylmalonate semialdehyde dehydrogenase gene.

### 2.8. Transcription Factors among the Core DEGs

Transcription factors (TFs) are important regulators of plant growth and development, and changes in their expression may lead to dramatic changes in gene transcription [44]. Studies have shown that TFs are the key junctions in the regulatory networks related to tapetum and pollen development [45]. A total of 23 of the 1287 core DEGs were annotated as TFs, belonging to six TF families, namely the bHLH (6), ERF (1), MYB (3), NAC (7), HD-Zip (2), and zinc finger protein (4) families. In particular, transcription factor ABORTED MICROSPORES (AMS), *bHLH089* and *bHLH091*, which are involved in pollen development, were significantly down-regulated in Ogura CMS (Figure 5D, Appendix A).

### 2.9. Quantification of Biochemical Indices Involved in Energy Metabolism

To clarify the effect of the down-regulation genes involved in energy metabolism, we measured the ATP content, ATPase activity, and H_2_O_2_ content in the anthers. The results showed that the ATP content and ATPase activity in Ogura CMS were significantly lower than those in the maintainer and restorer lines, but excessive H_2_O_2_ accumulated in Ogura CMS (Figure 6). Therefore, we speculate that the decrease in ATP content and the excessive accumulation of ROS caused the early degradation of the tapetum, and ultimately led to pollen abortion.

## 3. Discussion

There were obvious differences in flower organ morphology and pollen viability between the fertile line and the Ogura CMS line in cabbage. These differences became dispensable after the introduction of the restorer gene into Ogura CMS (Figure 1 and Figure 2). We detected the differences in gene expression among the three lines through full-length transcriptome sequencing and screened 1287 core DEGs through their shared genes. These DEGs had the same expression trend in the maintainer line and restorer line, but they were significantly up- or down-regulated in the Ogura CMS line, suggesting that these 1287 core DEGs may be closely related to pollen abortion (Figure 3). GO enrichment analysis showed that these DEGs are involved in lipid catalytic processes, pectin catalytic processes, proteolysis, the response to hydrogen peroxide, oxidoreductase activity, the pollen coat, and cell wall biology (Figure 4).

Pollen development is a complex process that involves the expression and regulation of many genes, and any change in a gene involved in this process may lead to pollen abortion [46]. The role of the tapetum in pollen development has been widely studied. The tapetum surrounding the microspore provides enzymes, nutrients, and pollen wall components during the continuous development of pollen [47]. Polygalacturonase (PG) is a ubiquitous cell wall protein in plants with pectin depolymerization activity. Silencing of *TaPG* in wheat leads to anther abnormalities, premature tapetal degradation, pollen abortion, and defects in pollen wall formation [48]. Pectin demethylation mediated by pectin methylesterases (PMEs) and PME inhibitors (PMEIs) is also crucial to morphological changes in the plant cell wall. The demethylation of pectin catalyzed by PMEs is arranged on the outer wall of the cell to make the cell wall hard, and the activity of PMEs can be specifically suppressed by PMEIs [49]. Callose plays an important role in the production of functional male gametophytes. In cotton, up-regulation of a pollen-specific protein (PSP231) activates callose biosynthesis and promotes pollen maturation, and silencing of PSP231 leads to abnormal pollen development and male sterility [50]. Transcription factor ABORTED MICROSPORES (AMS) is the main regulator of sporopollenin biosynthesis and pollen wall formation in *Arabidopsis*. *ams* mutants showed defective microspore release and a lack of sporopollenin deposition [51,52]. In *Arabidopsis*, *bHLH089* and *bHLH091* double mutants exhibit defective anther phenotypes, such as abnormal tapetal morphology, delayed callose degeneration, and aborted pollen development [53]. In this study, transcriptome analysis showed that the genes involved in tapetal and microspore development, such as PG, pectinesterase, pollen coat protein, callose synthase, GDSL esterase/lipase and cellulose synthase A subunit, AMS, *bHLH089,* and *bHLH091,* were significantly down-regulated in the Ogura CMS line. At the same time, cytological observations of microspore development showed the effects to phenotype due to the down-regulation of these genes. We observed that the tapetal cells in the Ogura CMS line were abnormally degraded at the tetrad stage and lost their original functions. Subsequently, the microspores stopped developing and eventually degraded together with the tapetum (Figure 2), which means that these genes play a crucial role in pollen abortion.

Pollen development is a highly energy-consuming process [54]. Mitochondria are the main sites for energy metabolism and redox [55]. The three major energy metabolic pathways in plants, oxidative phosphorylation, glycolysis, and the TCA cycle, are all related to the mitochondria. Most of the energy generated in the mitochondria is stored in the form of ATP for plant growth and development [56]. Therefore, numerous enzyme and protein complexes in the mitochondria are responsible for the synthesis and release of ATP, and mutation or activity reduction of these complex subunits may lead to an imbalance in energy metabolism, eventually causing male sterility [22]. NADH dehydrogenase (complex I) and cytochrome c oxidase (complex IV) are the first and last protein complexes in the respiratory electron transport chain, respectively, and play key roles in oxidative phosphorylation [57]. Inhibition of the activity of mitochondrial complex III in rice HL-CMS results in a decrease in ATP concentration and an increase in reactive oxygen species (ROS) content [25]. ATP synthetase (complex V) participates in the final and critical steps of ATP synthesis by converting the proton gradient force into chemical energy [44,58]. Previous studies have shown that many CMS-related genes are involved in the modification of ATP synthase subunits, such as *atp6* in *B. napus*, *atpA* in sunflower, *atp8* in radish, and *atp9* in petunia [59]. The decreases in ATP synthase activity and ATP content affect pollen development in CMS plants [60,61]. In this study, 20 genes involved in oxidative phosphorylation were significantly down-regulated in the Ogura CMS line, including seven genes encoding the ATPase subunit, four genes encoding the NADH dehydrogenase α subcomplex subunit, and two genes encoding the cytochrome c oxidase subunit. In addition, four key enzyme genes involved in glycolysis and the TCA cycle were down-regulated (Figure 5B). Meanwhile, the determination of biochemical indices related to energy metabolism proved that the ATP content and ATPase activity in Ogura CMS were significantly lower than those in the maintainer and restorer lines, but the H_2_O_2_ content was significantly increased (Figure 6). Studies have shown that the decrease in ATP synthase activity may lead to excessive accumulation of protons in the mitochondrial inner membrane, ultimately causing an explosion of ROS [62]. As important signals in plants, ROS play a dual role in inducing cell death and controlling various basic processes [63]. Excessive ROS can lead to lipid peroxidation, protein and DNA damage, increased cell permeability, and even abnormal tapetal PCD and pollen abortion [64,65,66].

In summary, based on cytological observations and transcriptome data analysis, we propose a relatively reliable network for the induction and regulation of cabbage Ogura CMS (Figure 7). During the tetrad stage of microspore development, the expression of key enzyme genes involved in glycolysis, the TCA cycle and oxidative phosphorylation is significantly down-regulated in Ogura CMS, leading to the inhibition of ATP synthesis and excessive accumulation of protons, thus triggering an explosion of ROS. The excessive ROS destroys the cell membrane wall, blocks the development of tapetum and gives rise to abnormal PCD in tapetum cells. The loss of tapetum function causes the microspore to stop developing due to the lack of carbohydrates, lipids, proteins, and other substances, and ultimately leads to pollen abortion.

## 4. Materials and Methods

### 4.1. Plant Materials

The plant materials used in this study were created and provided by the Institute of Vegetables and Flowers, Chinese Academy of Agricultural Sciences (IVF-CAAS). 19–616 is a normal fertile cabbage material, 19–2167 is an Ogura CMS material containing the sterility gene *orf138*, and FR2202 is a restorer line containing both the sterility gene *orf138* and the restoration gene *Rfo*. The lines had the same genetic background and were created through multigeneration backcrossing. The three materials were planted in the experimental base of IVF-CAAS in autumn 2019 and transplanted to the cold frame for vernalization in winter. In April of the following year, flower buds at the tetrad stage (2–3 mm) were collected, quickly frozen in liquid nitrogen, and then stored in a −80 °C freezer for transcriptome sequencing and fluorescence quantitative expression analysis [44].

### 4.2. Observation of Pollen Viability

The pollen viability of the three materials was observed by Alexander staining. At the full flowering stage, the newly opened flowers of the same day were removed, the petals and stamens were removed, the pollen was spread on a slide, and 2–3 drops of Alexander staining solution were added and mixed thoroughly. The slide was covered immediately, and the pollen was dyed for 3–6 h. Finally, the slides were observed under the microscope immediately after absorbing the excess liquid [67].

### 4.3. Library Construction and ONT Sequencing

The experimental process was performed according to the standard protocol provided by Oxford Nanopore Technologies (ONT). It included the following main steps: (1) extraction of total RNA; (2) NanoDrop analysis, Qubit analysis, agarose gel electrophoresis, or Agilent 2100 analysis for quality detection; (3) reverse transcription of the target mRNA using Oligo DT as a primer; (4) amplification of full-length cDNA by low-cycle PCR; (5) addition of a sequencing adapter (including motor protein); and (6) sequencing with a FLO-PRO002 chip.

After completion of library construction, the library with a certain concentration and volume was added to the flow cell, and the flow cell was transferred to an Oxford Nanopore PromethION sequencer for real-time single-molecule sequencing.

### 4.4. Sequencing Data Processing and Full-Length Transcript Identification

Clean data were obtained for subsequent analysis after filtering out low-quality reads and removing adapters from the raw FASTQ data. Linker sequences, sequences with quality values less than 7, and sequences with lengths less than 50 bp were filtered out.

Pychopper (version: 2.4.0; parameters: -Q 7 -z 50) was used to identify full-length sequences in the valid sequencing data. Pinfish (version: 0.1.0; parameter: default) was used to quickly construct a nonredundant transcript set for the full-length sequence. Minmap2 software (Version: 2.17-r941; parameters: -ax splice -uf -k14) was used to align the full-length sequences to the reference genome and obtain the BAM file. Then, the spliced_BAM2gff program was used to convert the BAM file to a GFF file, and the consistency sequence was obtained after clustering, removal of redundant reads, and correction using cluster_gff, collapse_partials and polish_clusters.

### 4.5. Quantitative Analysis and Screening of DEGs

The transcripts per kilobase million (TPM) value was used as the criterion for the gene expression level. For a single gene, the read count value was divided by the length of the gene (in kilobases) to obtain the reads per kilobase (RPK) coverage. All RPK values in the sample were calculated and then divided by 1,000,000 to obtain the per million scaling factor. The RPK value was divided by the per million scaling factor to obtain the final TPM value.

Differential expression analysis was performed according to the TPM values of transcripts in different samples. The software used for differential expression analysis was DESeq2, and the screening threshold was *padj* < 0.05 and |log_2_(fold change)| ≥ 1.

### 4.6. Annotation and Enrichment Analysis of DEGs

The identified differentially expressed transcripts were functionally annotated using seven databases, including the Nr, Pfam, eggNOG, UniProt, KEGG, GO, and COG databases.

Gene Ontology (GO) is an international standard classification system of gene functions. GO annotation information was simplified to obtain GOslim classifications, and the functions of transcripts were classified according to the cellular component (CC), molecular function (MF), and biological process (BP) categories. After summarization and statistical analysis, the second classification of GOslim with the most annotations under each classification was selected for drawing.

Kyoto Encyclopedia of Genes and Genomes (KEGG) is the main public database of the metabolic pathways and signal transduction pathways. Transcript sequences were annotated using the KEGG database and classified according to the KEGG metabolic pathways in which they were involved.

### 4.7. Real-Time Quantitative PCR Analysis

Ten DEGs were selected randomly for real-time quantitative PCR (qRT–PCR) to validate the results of the RNA-seq data. The RNA samples for qRT–PCR were identical to those used for RNA-seq. The PCR system was 20 μL, containing 10 μL of 2 × SYBR Green *Pro Taq* HS Premix (Code No. AG11701), 0.5 μL of forward and reverse primers, 7 μL of double-distilled water, and 2 μL of cDNA. Three technical replicates were performed for each sample. The expression level of each DEG was calculated using the 2^−ΔΔCt^ method with *Actin* as an internal reference gene. Primers were designed using PRIMER 5 (Appendix A).

### 4.8. Detection of Biochemical Indices Related to Energy Metabolism

Anthers were collected (with three biological replicates) at the tetrad stage (2–3 mm). An Enhanced ATP Assay Kit (S0027, Beyotime, Jiangsu, China) was used to measure the ATP content. CheKine™ Micro Ca^2+^/Mg^2+^-ATPase Activity Assay Kit (KTB1810, Abbkine, Wuhan, China) was used to measure the ATPase activity. The Hydrogen Peroxide (H_2_O_2_) Content Detection Kit (BC3595, Solarbio, Beijing, China) was used to measure the H_2_O_2_ content. The specific operation steps and calculation methods were conducted according to the kit instructions. The significance between the three materials was determined by Tukey’s Multiple Comparison Test (*p* < 0.05).

## Figures and Tables

**Figure 1 ijms-24-06703-f001:**
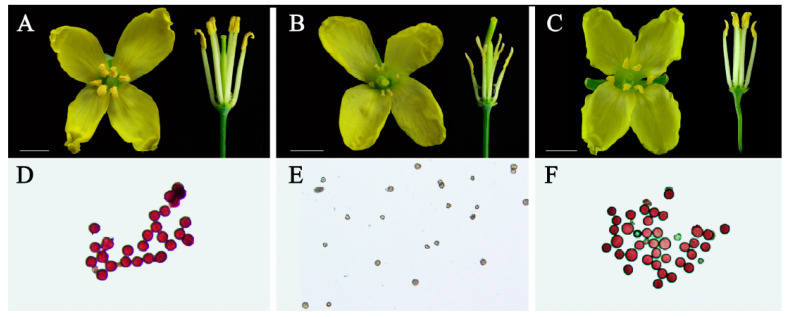
Flower morphology and pollen viability of the maintainer line, Ogura CMS line, and restorer line. (**A**,**D**) Maintainer line 19–616. (**B**,**E**) Ogura CMS 19–2167. (**C**,**F**) Restorer line FR2202. Bar = 0.5 cm.

**Figure 2 ijms-24-06703-f002:**
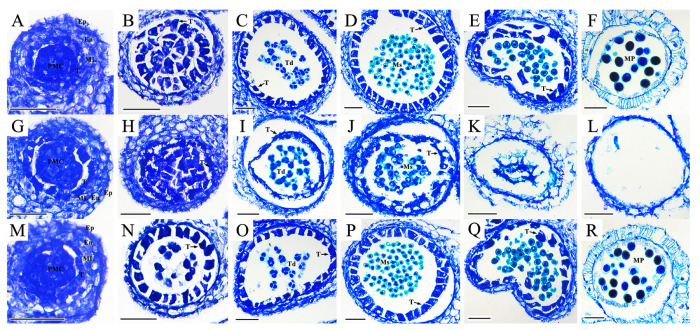
Cytological observation of anthers in the maintainer line, Ogura CMS line, and restorer line. (**A**–**F**) Maintainer line 19–616. (**G**–**L**) Ogura CMS 19–2167. (**M**–**R**) Restorer line FR2202. (**A**,**G**,**M**) Microsporocyte stage. (**B**,**H**,**N**) Meiotic stage. (**C**,**I**,**O**) Tetrad stage. (**D**,**J**,**P**) Uninucleate stage. (**E**,**K**,**Q**) Bicellular microspore stage to trinucleate microspore stage. (**F**,**L**,**R**) Mature pollen stage. Bar = 50 µm. PMC, pollen mother cell; Ep, epidermis; En, endothecium; ML, middle layer; T, tapetum; Td, tetrad; Ms, microspore; MP, mature pollen.

**Figure 3 ijms-24-06703-f003:**
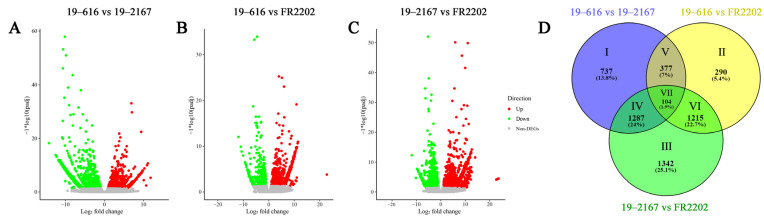
Differential expression analysis of nuclear genes. (**A**–**C**) The pairwise comparison between 19–616, 19–2167 and FR2202. (**D**) Analysis of the shared DEGs in 19–616, 19–2167 and FR2202.

**Figure 4 ijms-24-06703-f004:**
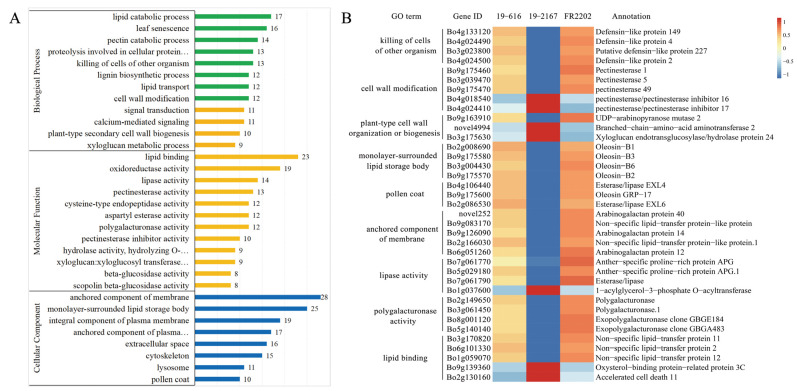
Gene ontology analysis of the core DEGs enriched in part IV. (**A**) The number of DEGs enriched in the three categories. (**B**) The expression trends and annotations of the core DEGs.

**Figure 5 ijms-24-06703-f005:**
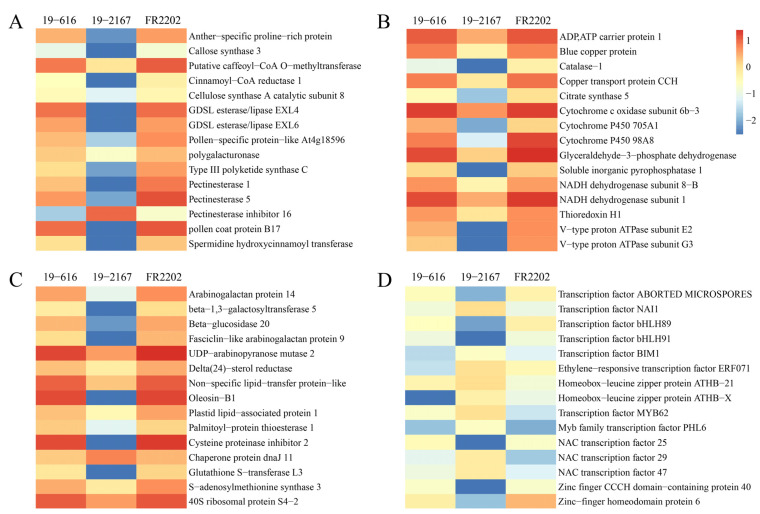
The expression trends and annotations of DEGs enriched in pollen development (**A**); energy metabolism (**B**); carbohydrate, lipid, and protein metabolism (**C**); and transcription factors (**D**).

**Figure 6 ijms-24-06703-f006:**
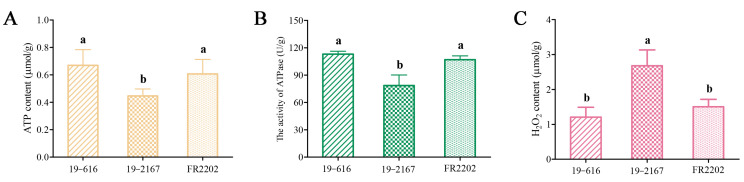
Mitochondrial biochemical indices among three lines. (**A**) ATP content. (**B**) ATPase activity. (**C**) H_2_O_2_ content. Significant differences between samples labeled with different Roman letters (a, b) were determined by Turkey’s tests. *p* < 0.05.

**Figure 7 ijms-24-06703-f007:**
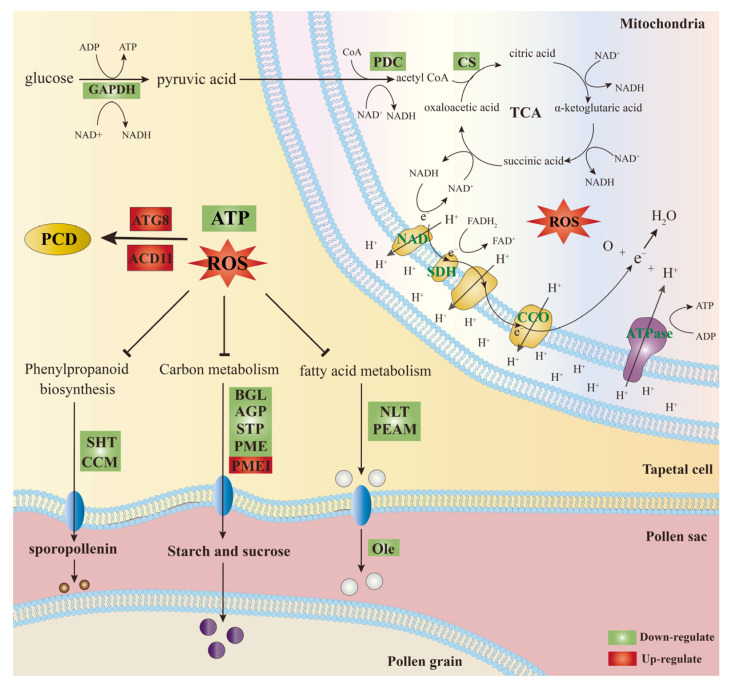
Potential transcriptome-mediated regulation network in cabbage Ogura CMS. PCD, programmed cell death; GAPDH, glyceraldehyde-3-phosphate dehydrogenase; PDC, pyruvate dehydrogenase; CS, citrate synthase; ND, NADH dehydrogenase; SDH, succinate dehydrogenase; CCO, cytochrome c oxidase; ATG8, autophagy-related protein 8; ACD11, accelerated cell death 11; SHT, spermidine hydroxycinnamoyl transferase; CCM, caffeoyl-CoA O-methyltransferase; BGL, Beta-glucosidase; AGP, arabinogalactan protein; STP, sugar transport protein; PME, pectinesterase; PMEI, Pectinesterase inhibitor; NLT, non-specific lipid-transfer protein; PEAM, phosphoethanolamine N-methyltransferase; Ole, Oleosin.

**Table 1 ijms-24-06703-t001:** Sequencing data and assembly results.

	19–616	19–2167	FR2202
BaseNum (Gb)	5.28	7.05	4.90
ReadNum	7,558,047	10,658,677	7,245,445
PassReads	7,557,374	10,657,713	7,244,891
Primers_found	6,429,226	9,214,366	6,313,210
N50	897	857	841
MeanLength	697.7	656.3	680.3
MaxLength	183,372	128,123	113,851
MeanQscore	10.3	10.4	10.7

Note: BaseNum: total number of bases; ReadNum: total number of reads. PassReads: the number of reads with the quality value greater than 7 and a length longer than 50 bp; Primers_found: the number of full-length sequences; N50: the length of N50; MeanLength: mean length of reads; MaxLength: the longest length of reads. MeanQscore: mean quality value.

## Data Availability

The datasets used and/or analyzed during the current study are available from the corresponding author upon reasonable request.

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
