# Peer review of "Comparative Transcriptome Analysis Reveals a Potential Regulatory Network for Ogura Cytoplasmic Male Sterility in Cabbage (Brassica oleracea L.)"

_ijms, 2023, doi:10.3390/ijms24076703_

Round 1
Reviewer 1 Report
The study of CMS is an important scientific task. The presented work "Comparative transcriptome analysis reveals a potential regulatory network for Ogura cytoplasmic male sterility in cabbage (Brassica oleracea L.)" aims to solve this problem in a single case of CMS in cabbage. The design of the experiment meets the stated goal, the methodology is adequate. The authors' conclusions are consistent with the expected results. However, I had a question regarding the openness of the described data: are the sequencing results in the public domain, for example, on the SRA servers (https://www.ncbi.nlm.nih.gov/sra). Also in the publication there are minor editorial flaws: Line 118: In the caption to figure 2, is there a typo in the notation C,I,L (maybe -- CIO)? Lines 144-155: Is a tabular data description necessary? Line 392: The sentence "The three materials were planted in the ..." is not entirely clear, it should be made clearer. I would also like to note that the resolution of the drawings is not the best, in particular - Figure 5.
Nevertheless, with the correction of the noted shortcomings, this article may well be published in this journal. And the shortcomings in themselves do not detract from the content component.
Author Response
Thank you very much for your suggestions on this manuscript. We admire you for your careful and professional inspection. We have made modifications according to your suggestions, and the following are replies one by one.
Point 1: Are the sequencing results in the public domain, for example, on the SRA servers (https://www.ncbi.nlm.nih.gov/sra).
Response 1: We are currently working on this task. Once the upload is successful, we will inform the editors as soon as possible.
Point 2: Line 118: In the caption to figure 2, is there a typo in the notation C,I,L (maybe -- CIO)?
Response 2: Yes, it was a typo and we have replaced the L with an O.
Point 3: Lines 144-155: Is a tabular data description necessary?
Response 3: We wanted to retain the table to show the detailed data and sequencing quality more visualized.
Point 4: Line 392: The sentence "The three materials were planted in the ..." is not entirely clear, it should be made clearer.
Response 4: We have supplemented this part.
Point 5: The resolution of the drawings is not the best, in particular Figure 5.
Response 5: We modified the Figure 5 and improved the resolution

Reviewer 2 Report
The work of the authors is devoted to a very important aspect - cytoplasmic male sterility. The work was carried out on several lines of cabbage, in which the maturation of pollen is maintained or its abartation occurs. Cabbage is an important agricultural crop, and the study of its male sterility is very important for breeding, as it is one of the main tools in hybridization in obtaining new varieties with valuable properties.
The work of the authors was carried out at a high methodological level and makes a significant contribution to understanding the mechanisms of genetic regulation of male sterility and the cell compartments involved in it.
At the same time, several minor shortcomings can be found in the work, which the authors need to correct before the work can be recommended for publication: 1) there are no Bars in Figure 1; 2) Figure 3K - the fragment does not contain tissue signatures, however, it is believed that at this stage the tapetum has not yet degraded and it can be distinguished; 3) it is not clear how the authors checked the reliability of the values in the experiment, in the methodology section there is no section on statistical data processing - it is necessary to add these data.
Author Response
Thank you very much for your suggestions on this manuscript. We admire you for your careful and professional inspection. We have made modifications according to your suggestions, and the following are replies one by one.
Point 1: There are no Bars in Figure 1.
Response 1: We have added Bars in Figure 1.
Point 2: Figure 3K - the fragment does not contain tissue signatures, however, it is believed that at this stage the tapetum has not yet degraded and it can be distinguished.
Response 2: For fertile line and restorer line, the tapetum did not degrade during this period, but for CMS, no obvious tapetum signatures have been observed, so we believe that it is in a degraded state.
Point 3: it is not clear how the authors checked the reliability of the values in the experiment, in the methodology section there is no section on statistical data processing - it is necessary to add these data.
Response 3: We have added the section on statistical data processing in 4.8.

Reviewer 3 Report
This study performed a systematic characterization investigate the microspore development process and gene expression changes during cabbage pollen development. The manuscript seeks to answer important questions. Transcriptome libraries were compared from flower buds of the cytoplasmic male sterile and its maintainer fertile lines to determine the source of sterility. It makes use of a variety of precise methods that are suitable for achieving the objectives set. A large amount of data has been generated during the characterization of the 1287 DEGs. From the sequencing data and the differential expression analysis, the male fertile and restorer lines show high similarity. Did you find cardinal differences? Can you provide some milestone of the restoring mechanism? Cannot the ROS eliminating enzymatic pathways play role in CMS or in restoration? Consider the omitting of Fig. 7, it is not informative enough, and takes too much space. The graphs are sufficiently detailed and easy to expound. Font size of the manuscript should be consolidated. The text contains occasional typos, but it is nevertheless easy to read. It uses scientific language appropriately. The literature used is fair enough.
Author Response
Thank you very much for your suggestions on this manuscript. We admire you for your careful and professional inspection. We have made modifications according to your suggestions, and the following are replies one by one.
Point 1: From the sequencing data and the differential expression analysis, the male fertile and restorer lines show high similarity. Did you find cardinal differences?
Response 1: In our study, a total of 1986 DEGs were identified between the maintainer line and the restorer line, but we have not yet conducted a detailed analysis of these genes
Point 2: Can you provide some milestone of the restoring mechanism?
Response 2: I'm sorry we haven't done much in-depth research on the mechanism of fertility recovery. However, for the fertility restoration of Ogura CMS, the article 'The radish Ogura fertility restorer impales translation correlation along with its corresponding CMS-causing mRNA' has made a breakthrough.
Point 3: Cannot the ROS eliminating enzymatic pathways play role in CMS or in restoration?
Response 3: We have not found significant role for ROS elimination enzymes in Ogura CMS
Point 4: Consider the omitting of Fig. 7, it is not informative enough, and takes too much space. The graphs are sufficiently detailed and easy to expound.
Response 4: We want to display the results more visualized through the schematic diagram, so we have made some supplements and modifications to Figure 7.
Point 5: Font size of the manuscript should be consolidated. The text contains occasional typos, but it is nevertheless easy to read.
Response 5: We have carefully reviewed the full text, and corrected the inappropriate fonts and typos.
